# Learners' Online Self-Regulated Learning Skills in Indonesia Open University: Implications for Policies and Practice

Harry B. Santoso [1], Rahayu Dwi Riyanti [2], Trini Prastati [2], FA. Triatmoko H. S. [3], Arie Susanty [4] and Min Yang [5],*

1 Faculty of Computer Science, Universitas Indonesia, Depok 16424, Indonesia; harrybs@cs.ui.ac.id
2 Faculty of Education and Teacher Training, Open University of Indonesia, Tangerang Selatan 15437, Indonesia; rahayudr@ecampus.ut.ac.id (R.D.R.); trini@ecampus.ut.ac.id (T.P.)
3 Directorate of Academic Development and Learning Resources, Universitas Indonesia, Depok 16424, Indonesia; fatrihs@ui.ac.id
4 Research and Development Division, SEAMOLEC, Tangerang Selatan 15418, Indonesia; susanty@seamolec.org
5 Department of Curriculum and Instruction, The Education University of Hong Kong, New Territories, NT, Hong Kong, China
* Correspondence: myang@eduhk.hk

**Abstract:** To succeed in online distance learning where students are physically separated from teachers and peers, students must develop self-regulated learning skills to effectively manage their learning process. This study examined how students with different demographic backgrounds adopt or fail to adopt self-regulated learning to engage in online distance learning. Survey data were collected from 295 students at Indonesia Open University. Although students' online learning self-efficacy and online self-regulated learning were above average, they reported low levels of confidence and abilities to seek help, develop task strategies, and allocate time for online learning. Their online self-regulated learning and online learning self-efficacy were significantly correlated. However, there were no significant relationships between online self-regulated learning and learning performance, and between online learning self-efficacy and learning performance. Furthermore, female and/or older students reported lower learning performance than male and younger students. Qualitative data from open-ended questions were analyzed to interpret the quantitative results. Recommendations for stakeholders of open universities were made to assist in students' improvement of self-regulated learning skills and to address equity issues in Indonesian online distance learning and similar contexts.

**Keywords:** open university; online distance learning; online self-regulated learning; online learning self-efficacy

## 1. Introduction

Indonesia is a nation with a vast geographical area spanning from Sabang in North Sumatera to Merauke in Papua and has more than 17,000 islands. Among the over 270 million citizens, only 30.85% of them have gained higher education qualifications [1]. Universitas Terbuka or Indonesia Open University (IOU) was established in 1984 to serve learners who would otherwise be unable to attend higher education due to various constraints, such as lack of funding, living in isolated or rural areas, or having full-time jobs [2]. Since 2016, IOU has started to offer fully online programs and is now moving to become a cyber-university [3].

One way to increase learners' access to higher education and enhance their quality of learning is by offering online distance learning [4]. A key feature of online distance learning is the flexibility that it offers, allowing students the freedom to learn without being by confined by time and place [5]. Learners can access courses at their convenience

through the online learning system, where they undertake individual and collaborative learning activities.

For students of online distance learning who are physically separated from their teachers and peers, it is essential that they develop online self-regulated learning skills to effectively manage their learning process and attain academic success [6]. Self-regulated learning refers to "the ways that learners systematically activate and sustain their cognitions, motivations, behaviors, and affects, toward the attainment of their goals" [7]. In both face-to-face and online learning environments, existing studies have reported positive relationships between self-regulated learning and learning outcomes [8,9] and between self-regulated learning and demographic variables, such as age and gender [10,11]. However, no existing studies have investigated such relationships among Indonesian online distance learning students. Given such research gap, this study investigated the following research questions through a survey with 295 IOU students:

*RQ 1*: What are students' demographic profiles of online self-efficacy, online self-regulated learning, and learning performance?
*RQ 2*: How are students' online self-efficacy, online self-regulated learning, and learning performance correlated?
*RQ 3*: How do students differ in their online self-efficacy, online self-regulated learning, and learning performance by demographic variables?
*RQ 4*: What are students' perceptions of their learning experiences, strategies, and challenges, and suggestions for learning support?

## 2. Self-Regulated Learning, Learning Self-Efficacy, and Learning Performance in Online Learning Environments

When learning in both face-to-face and online environments, learners need to self-regulate learning by setting goals for tasks, constructing learning schedules to meet the goals, and staying on track to complete course work on time. Self-regulated learning is based on three assumptions [12]. First, the learner is an active participant in the learning process. Second, the learner has the potential for controlling various aspects of learning, including cognition (checking and assessing learning progress, correcting errors, and adjusting strategies), motivation (self-encouragement to complete a task or goal, which involves self-efficacy beliefs, task value beliefs, and goal orientation; see [13] and [14]), behavior (activities that can be observed during learning), and the environment (e.g., learning space, communication tools, rules of the class). Third, the learner is able to self-evaluate learning (comparing current progress with task goals, success criteria, or standards) to identify learning gaps and modify task strategies. Concurring with such assumptions, Schunk et al. [15] proposed six dimensions of self-regulated learning, which are (1) Goals and self-efficacy, (2) Strategy use or routinized performance, (3) Time management, (4) Self-observation, self-judgment, and self-reaction, (5) Environmental structuring, and (6) Selective help seeking.

Specific to online learning, previous studies found evidence on the relationships between students' self-regulated learning, self-efficacy, demographic backgrounds, and learning achievement [8,16]. For example, in Wang et al.'s (2013) study with 256 online learning students, demographic variables and self-regulated learning predicted learning performance and course satisfaction through motivation and self-efficacy. Self-efficacy is defined as learners' "beliefs in one's capabilities to organize and execute the courses of action required to produce given attainments" [17]. Students' self-efficacy beliefs may differ in online learning and face-to-face learning environments. For example, in Levterova-Gadjalova and Tsokov's study [18], students reported lower levels of self-efficacy when learning in online learning courses compared to learning face-to-face.

We set out to examine the relationships between Indonesian distance learning students' online self-regulated learning, online learning self-efficacy, and learning performance, because no previous studies have been conducted to examine these relationships among this student population. Most existing studies examined the correlation between online self-regulated learning and mathematics skills [19] or spatial abilities [20]. Existing Indonesian

studies on online learning self-efficacy have only focused on its relationships with online learning behavior, for example, frequency of accessing e-learning websites [21] and course satisfaction [22] among traditional university students. By conducting this study, we aim to fill the above research gaps.

## 3. Research Context

In Indonesia, distance education in higher education is governed by the Ministry of Education and Culture in Ministerial Regulation No. 109/2013. The IOU adopts a distance education system where learning processes in all educational programs are carried out at a distance using learning media. IOU students apply an independent learning system. Depending on their own initiative, desire, or interest, students undertake learning activities individually or in groups, such as in a tutorial group. Students learn not only from the provided learning materials in various learning media, but also other learning resources, such as libraries, as well as television and radio broadcasts. If students encounter difficulties, they are encouraged to ask lecturers for assistance during lectures or by other means such as emails.

In 2021, there were 310,974 IOU students from different backgrounds [23]. The largest groups are teachers (41%), private sector workers (20.07%), and unemployed people (13.43%). Most of them are women (64.4%), of whom 40.6% are below 25 years of age. The majority of the students come from Java Island (43.16%), Sumatera (28.24%), and Kalimantan (11.91%). Some of them also come from abroad, such as Malaysia (40.82%), Hong Kong (10.24%), and Taiwan (8.65%).

## 4. Methods

### 4.1. Participants and Procedures

We used convenience sampling in recruiting participants. The online survey was administered using Qualtrics. The researchers collaborated with the coordinator of academic administrative office at IOU. The survey collected quantitative and qualitative data from new learners (2019–2020 entrants) at IOU ($n$ = 295). The online survey stressed confidentiality of participants' private information and their right to withdraw from the survey without any adverse consequences.

Among the participants, 44.41% were males, 54.58% were females, and 1.02% did not specify their gender. Moreover, 73.56% of the participants were between 20 and 30 years old, 17.97% were between 30 and 40 years old, and just 8.47% were either under 20 years old or above 40 years old. Lastly, 21.02% of them registered at IOU before 2019, 30.85% registered in 2019, and 48.14% of the participants registered in 2020. The participants were from seven bachelor's programs of four faculties (see Table 1).

**Table 1.** Students' Faculties and Program of Studies.

| Faculties | Programs | % | Number of Participants |
|---|---|---|---|
| Faculty of Teacher Training | Educational Technology | 3.73% | 11 |
| Faculty of Economics | Development Economics | 7.46% | 22 |
| | Management | 52.20% | 154 |
| Faculty of Science and Technology | Mathematics | 2.37% | 7 |
| | Urban and Regional Planning | 4.07% | 12 |
| Faculty of Law, Social and Political Sciences | State Administration Science Study | 9.49% | 28 |
| | Communication Studies | 20.68% | 61 |
| Total | | 100% | 295 |

### 4.2. Measurements

Quantitative data were obtained using three questionnaires to captures students' demographic profiles, online self-efficacy, and online self-regulated learning skills. Students' performance in assignments was used to represent their learning performance. For the

qualitative data, themes were constructed from the answers to open-ended questions. These questions capture students' perspectives about personal goals, positive learning experiences, learning challenges, learning strategies, and need for learning support in online learning context.

The Self-Efficacy Questionnaire for Online Learning (SeQOL) is a 25-item questionnaire using 11-point Likert scale (0 = cannot do at all, 5 = moderately confident can do, 10 = highly confident can do) adapted to Indonesian language from Tsai et al. (2020). It consists of five dimensions including self-efficacy to (1) complete an online course, (2) interact socially with classmates, (3) handle tools in a CMS, (4) interact with instructors in an online course, and (5) interact with classmates for academic purposes.

SeQoL is a valid and reliable instrument to measure online learning self-efficacy and the overall scores were correlated with general self-efficacy, learning satisfaction, and expected grades [24]. The internal consistency was measured by Cronbach's $\alpha$ and was found to be 0.976. Cronbach's $\alpha$ for the SeQOL questionnaire dimensions is provided in Table 2 below. This result shows that the Indonesian version of SeQOL is consistent in measuring self-efficacy in online learning.

**Table 2.** SeQOL internal consistency.

| Scale Reliability Statistics | Cronbach's $\alpha$ | Sample Item |
|---|---|---|
| Scale | 0.976 | |
| SE to complete an online course | 0.952 | Willing to face challenges |
| SE to interact socially with classmates | 0.911 | Pay attention to other students' social actions |
| SE to handle tools | 0.839 | Send email to others with or without attached |
| SE to interact with instructors | 0.947 | Clearly ask my questions to instructor |
| SE to interact with classmates for academic purposes | 0.925 | Actively participate in online discussions |

The validity of SeQOL is measured using criterion-related procedure with the students' self-regulated learning skills, since self-efficacy and self-regulated learning are related [25]. Pearson correlation matrix at Table 3 showed a significant correlation between self-efficacy in online learning and skills to manage online learning independently ($r = 0.572$, $p < 0.01$). This result shows that the Indonesian version of SeQOL is a valid instrument measuring self-efficacy in online learning.

**Table 3.** Correlation matrix between SeQOL and OSRL.

| | | SeQOL Score | OSRL Score |
|---|---|---|---|
| SeQOL Score | Pearson's $r$ | — | |
| | $p$-value | — | |
| | N | — | |
| OSRL Score | Pearson's $r$ | 0.572 *** | — |
| | $p$-value | <0.001 | — |
| | N | 295 | — |

Note. *** $p < 0.001$.

The Online Self-Regulated Learning Questionnaire is a 24-item questionnaire using 6-point Likert scale (1 = very inappropriate, 6 = very appropriate) developed by Arbiyah and Triatmoko [26]. The internal consistency is measured by Cronbach's $\alpha$ and was found to be 0.905. The instrument validity is measured using a criterion-related procedure with the students' performance, which resulted in significant correlation ($r = 0.36$, los 0.05) [26]. It consists of six dimensions, which are (1) environment structuring, (2) goal setting, (3) help-seeking, (4) self-evaluation, (5) task strategies, and (6) time management. From this research, the internal consistency was measured by Cronbach's $\alpha$ and was found to be 0.923. Cronbach's $\alpha$ for the online self-regulated learning questionnaire dimensions is provided at Table 4 below. This result shows that the instrument is consistent in measuring self-regulated learning in online learning.

**Table 4.** The internal consistency of the OSRL Questionnaire Scale Reliability Statistics.

| | Cronbach's $\alpha$ | Sample Item |
|---|---|---|
| Scale | 0.923 | |
| Environment structuring | 0.777 | I can choose the right study location in online learning to avoid too many distractions. |
| Goal Setting | 0.876 | I set targets for my assignments in online learning. |
| Help-seeking | 0.853 | I know who to ask if I encounter difficulties while studying in online learning. |
| Self-evaluation | 0.852 | I evaluate the extent of my understanding of the learning materials in the online learning that I take. |
| Task Strategies | 0.504 | I do not have a specific strategy for completing assignments in online learning. |
| Time management | 0.608 | I allocate additional study time for online learning because I know that online learning requires good time management. |

In this study, data were also gathered in qualitative format; participants' responses to open-ended questions were analyzed to provide additional insights into students' learning in online context. These are the open-ended questions that ask them about their personal goals, learning experiences, learning strategies, learning challenges, and suggested learning support.

### 4.3. Data Analysis

Quantitative data gathered were analyzed using different statistical analyses (i.e., descriptive statistics analysis, Pearson's correlations analysis, Cronbach's $\alpha$ reliability analysis). To provide demographic profiles and description of online self-efficacy and online self-regulated learning, descriptive statistics were used. Pearson's correlations [27] were applied to analyze the correlations among online self-efficacy, online self-regulated learning, and learning performance. The analyses were conducted by also considering the students' demographic profiles.

In addition, qualitative data were analyzed using the Python programming language with the Latent Dirichlet Allocation algorithm [28]. This algorithm was chosen for conducting topic modeling. Before processing the data with LDA, a pre-processing stage was carried out in the form of data cleaning. The steps for cleaning the data included changing the uppercase/capital letters to lowercase letters, then removing the punctuation marks and eliminating one character. The LDA model was evaluated by looking at the graph of the coherence score versus the number of topics. The higher the score, the better the cluster. After that, each cluster was analyzed qualitatively to summarize the main themes emerging from the clusters.

### 5. Results

*5.1. Addressing RQ 1: Students' Demographic Profiles, Levels of Online Self-Efficacy, Online Self-Regulated Learning Skills, and Learning Performance*

5.1.1. Demographic Profiles and Learning Performance

The demographic profiles provide an overview of the experiences of the respondents in this study with previous online learning and how they used learning technologies and internet access to support their online learning activities (see Tables 5–8). Around half of the students had ICT training and prior online learning experience; students who had such experience were likely to have a higher self-efficacy and self-regulation according to previous studies (e.g., [29,30]). Most students owned two or more devices, the majority of them chose to use mobile devices such as laptops and tablets for engaging in online learning, and most of them frequently accessed the internet in each week. These results show that the students were able to learn anywhere and anytime by using mobile devices,

which allowed them to flexibly engage in online learning. As the students were grouped by age (under 30 years old and 30+ years old), there are differences in the score between age group. The 30+ years old student had higher score in OSRL and SeQOL, but a lower final grade (Table 9).

**Table 5.** Experience with online learning.

| Experience with Online Learning | % | Frequency |
|---|---|---|
| Have attended any courses such as MOOC and or short courses, before becoming an UT student | 41.93% | 135 |
| Have attended training related to ICT skills | 58.07% | 187 |
| Total | 100% | 322 |

**Table 6.** Number of devices used each day while the students' study.

| Number of Devices | % | Frequency |
|---|---|---|
| 1 device | 22.71% | 67 |
| 2 devices | 65.42% | 193 |
| More than 2 devices | 11.86% | 35 |
| Total | 100% | 295 |

**Table 7.** Time allocation for using the Internet in a week.

| Time Allocation | % | Frequency |
|---|---|---|
| Between 1–3 h | 8.47% | 25 |
| Between 3–6 h | 25.08% | 74 |
| More than 6 h | 66.44% | 196 |
| Total | 100% | 295 |

**Table 8.** Types of devices students use when participating in online distance learning.

| Types of Devices | % | Frequency |
|---|---|---|
| PC | 11.60% | 55 |
| Laptop | 52.11% | 247 |
| Tablet | 4.01% | 19 |
| Smartphones | 32.28% | 153 |
| Total | 100% | 474 |

**Table 9.** Descriptive Statistics of course grade, SeQOL, and OSRL divided by age.

| | Group Descriptives | | | | | |
|---|---|---|---|---|---|---|
| | **Group** | **N** | **Mean** | **Median** | **SD** | **SE** |
| OSRL Score | under 30 years | 222 | 107.5 | 111.0 | 23.6 | 1.58 |
| | 30+ years | 73 | 110.1 | 111.0 | 19.4 | 2.27 |
| SeQOL Score | under 30 years | 222 | 159.3 | 171.5 | 52.8 | 3.54 |
| | 30+ years | 73 | 173.8 | 185.0 | 48.7 | 5.71 |
| Grade | under 30 years | 178 | 74.4 | 82.0 | 23.9 | 1.79 |
| | 30+ years | 61 | 68.2 | 80.0 | 27.5 | 3.52 |

### 5.1.2. Descriptive Statistics of Online Learning Self-Efficacy

The Self-Efficacy Questionnaire for Online Learning (SeQOL) is a 25-item questionnaire using 11-point Likert-scale (0 = cannot do at all, 5 = moderately confident can do, 10 = highly confident can do).

Table 10 above shows that the means of all questionnaire items of online self-efficacy were above the mid-point score of the SeQOL's Likert scale (i.e., 5). Some statements from the Self-Efficacy Questionnaire for Online Learning have a higher mean than 7.00 (out of 10.00).

**Table 10.** Descriptive Statistics of SeQOL.

| Statement | N | Missing | Mean | Median | SD | Variance | Min. | Max. |
|---|---|---|---|---|---|---|---|---|
| Willing to face challenges | 292 | 3 | 6.91 | 8.00 | 2.54 | 6.44 | 0 | 10 |
| Create a plan to complete the given assignments | 292 | 3 | 6.93 | 7.00 | 2.36 | 5.59 | 0 | 10 |
| Willingly adapt my learning styles to meet course expectations | 292 | 3 | 7.18 | 8.00 | 2.35 | 5.54 | 1 | 10 |
| Understand complex concepts | 292 | 3 | 6.52 | 7.00 | 2.22 | 4.91 | 0 | 10 |
| Keep up with course schedule | 292 | 3 | 7.44 | 8.00 | 2.41 | 5.79 | 0 | 10 |
| Evaluate assignments according to the criteria provided by the instructor | 292 | 3 | 6.86 | 7.00 | 2.39 | 5.69 | 0 | 10 |
| Complete an online course with a good grade | 292 | 3 | 7.39 | 8.00 | 2.35 | 5.50 | 0 | 10 |
| Pay attention to other students' social actions | 292 | 3 | 6.03 | 6.00 | 2.25 | 5.07 | 0 | 10 |
| Initiate social interaction with classmates | 292 | 3 | 6.09 | 6.00 | 2.58 | 6.64 | 0 | 10 |
| Apply different social interaction skills depending on situations | 292 | 3 | 6.42 | 7.00 | 2.30 | 5.31 | 0 | 10 |
| Develop friendship with my classmates | 292 | 3 | 6.07 | 6.00 | 2.70 | 7.26 | 0 | 10 |
| Send email to others with or without attachment | 292 | 3 | 5.42 | 5.00 | 2.74 | 7.53 | 0 | 10 |
| Reply to others' messages in a discussion board | 292 | 3 | 6.24 | 6.00 | 2.52 | 6.37 | 0 | 10 |
| Post a new message in a discussion board | 292 | 3 | 6.75 | 7.00 | 2.42 | 5.86 | 0 | 10 |
| Clearly ask my questions to instructor | 292 | 3 | 6.66 | 7.00 | 2.57 | 6.60 | 0 | 10 |
| Seek help from instructor when needed | 292 | 3 | 6.62 | 7.00 | 2.61 | 6.82 | 0 | 10 |
| Inform the instructor in a timely manner when unexpected situations arise | 292 | 3 | 6.52 | 7.00 | 2.65 | 7.01 | 0 | 10 |
| Initiate discussions with the instructor | 292 | 3 | 6.37 | 7.00 | 2.55 | 6.50 | 0 | 10 |
| Express my opinions to instructor respectfully | 292 | 3 | 7.12 | 8.00 | 2.49 | 6.18 | 0 | 10 |
| Actively participate in online discussions | 292 | 3 | 7.28 | 8.00 | 2.48 | 6.15 | 0 | 10 |
| Effectively communicate with my classmates | 292 | 3 | 6.28 | 7.00 | 2.63 | 6.92 | 0 | 10 |
| Respond to other students in a timely manner | 292 | 3 | 6.21 | 7.00 | 2.52 | 6.33 | 0 | 10 |
| Request help from others when needed | 292 | 3 | 5.89 | 6.00 | 2.62 | 6.88 | 0 | 10 |
| Express my opinions to other students respectfully | 292 | 3 | 6.69 | 7.00 | 2.53 | 6.42 | 0 | 10 |
| Provide help to other students when assistance is needed | 292 | 3 | 6.66 | 7.00 | 2.55 | 6.50 | 0 | 10 |

### 5.1.3. Descriptive Statistics of Online Self-Regulated Learning

The Online Self-Regulated Learning (OSRL) questionnaire was answered by students using a 6-point Likert scale of 1 (Strongly Disagree) to 6 (Strongly Appropriate) for each item, as a reflection of students' learning experience in the online class (see Table 11 below).

**Table 11.** Descriptive Statistics of OSRL.

| Statement | N | Missing | Mean | Median | SD | Variance | Min. | Max. |
|---|---|---|---|---|---|---|---|---|
| I can choose the right study location in online learning to avoid too many distractions. | 291 | 4 | 5.32 | 6 | 0.950 | 0.902 | 1 | 6 |
| I close all tabs or windows that are not related to learning material while taking lessons in online learning. | 291 | 4 | 4.68 | 5 | 1.333 | 1.778 | 1 | 6 |
| I know where I can study most effectively for online learning. | 291 | 4 | 5.30 | 6 | 0.959 | 0.919 | 1 | 6 |
| I choose study times that have the least amount of distractions in online learning. | 291 | 4 | 5.17 | 5 | 1.055 | 1.113 | 1 | 6 |
| I set targets for my assignments in online learning. | 291 | 4 | 5.14 | 5 | 1.069 | 1.142 | 1 | 6 |
| I set short-term goals (daily or weekly) that I want to achieve in my online learning. | 291 | 4 | 4.98 | 5 | 1.149 | 1.320 | 1 | 6 |
| I set high standards for my learning in online learning. | 291 | 4 | 4.92 | 5 | 1.088 | 1.183 | 1 | 6 |
| I set a long-term goal (monthly or semester) that I want to achieve in my online course. | 291 | 4 | 4.89 | 5 | 1.101 | 1.213 | 1 | 6 |
| I know who to ask if I encounter difficulties while studying in online learning. | 291 | 4 | 4.69 | 5 | 1.332 | 1.774 | 1 | 6 |
| I asked other people who had attended online learning about how to study effectively in online learning. | 291 | 4 | 4.33 | 5 | 1.495 | 2.234 | 1 | 6 |
| I contacted a classmate in an online learning subject when I had difficulties in learning. | 291 | 4 | 4.08 | 4 | 1.570 | 2.466 | 1 | 6 |
| I share problems in online learning with my classmates, so we know what problems we have together and how to solve them. | 291 | 4 | 4.08 | 4 | 1.610 | 2.593 | 1 | 6 |
| I evaluate the extent of my understanding of the learning materials in the online learning that I take. | 291 | 4 | 4.74 | 5 | 1.137 | 1.292 | 1 | 6 |
| I communicate with my classmates to find out if what I understand is different from what they understand. | 291 | 4 | 4.13 | 4 | 1.507 | 2.270 | 1 | 6 |
| I evaluate whether the learning strategy I use is able to achieve the target I have set at the beginning of the online learning. | 291 | 4 | 4.64 | 5 | 1.261 | 1.590 | 1 | 6 |
| In the middle of the semester, I reflected again on whether the learning strategies I used in online learning were effective. | 291 | 4 | 4.66 | 5 | 1.179 | 1.390 | 1 | 6 |
| I do not have a specific strategy for completing assignments in online learning. (REVERSED) * | 291 | 4 | 3.49 | 3 | 1.697 | 2.878 | 1 | 6 |
| I make a strategy for doing assignments in online learning. | 291 | 4 | 4.76 | 5 | 1.212 | 1.468 | 1 | 6 |
| I prepare questions that I will ask before joining discussion forums or chat rooms. | 291 | 4 | 4.13 | 4 | 1.407 | 1.980 | 1 | 6 |
| I do additional things in online learning other than those assigned to me to master the learning material. | 291 | 4 | 4.24 | 4 | 1.345 | 1.809 | 1 | 6 |
| I allocate additional study time for online learning because I know that online learning requires good time management. | 291 | 4 | 4.66 | 5 | 1.258 | 1.583 | 1 | 6 |
| There is no specific time that I allocate to study in online learning. (REVERSED) * | 291 | 4 | 3.54 | 3 | 1.623 | 2.636 | 1 | 6 |
| I determine the number of study hours that I will allocate each week for online learning. | 291 | 4 | 4.59 | 5 | 1.215 | 1.476 | 1 | 6 |
| I set the same schedule every day or every week to study in online learning. | 291 | 4 | 4.52 | 5 | 1.322 | 1.747 | 1 | 6 |

* Note: Reverse questionnaire item.

Table 11 shows that the means of questionnaire items of online self-regulated learning were above the mid-point of the OSRL's Likert scale (i.e., between 3 and 4). Some statements from the Online Self-Regulated Learning (OSRL) had mean scores higher than 5.00 (out of 6.00): "I can choose the right study location in online learning to avoid too many distractions (*M* = 5.23); "I know where I can study most effectively for online learning" (*M* = 5.30); "I choose study times that have the least amount of distractions in online learning" (*M* = 5.17); "I set targets for my assignments in online learning" (*M* = 5.14). In contrast, two statements had means below 4.00 (out of 6.00): "I do not have a specific strategy for completing assignments in online learning" (*M* = 3.49) and "There is no specific time that I allocate to study in online learning" (*M* = 3.54), indicating their need for support to enhance skills for time management and task strategy. Two items related to seeking help and feedback from classmates had relatively low means as well: "I contacted a classmate in an online learning when I had difficulties in learning" (*M* = 4.08) and "I share problems in online learning with my classmates, so we know what problems we have together and how to solve them" (*M* = 4.08). Help seeking to obtain feedback is one of the important ways of improving students' learning performance, which can be enhanced by enhancing social interaction in the online learning environment [31].

### 5.2. Addressing RQ 2: The Correlations among Students' Online Self-Efficacy, Online Self-Regulated Learning, and Learning Performance

#### 5.2.1. Correlations between Online Self-Efficacy and Online Self-Regulated Learning

Pearson correlation showed a significant correlation between self-efficacy in online learning and skills to manage online learning independently (*r* = 0.572, *p* < 0.001). Students who reported high self-efficacy for online learning also rated their online self-regulated learning to be high, and vice versa (see Table 12). The findings of this study are in line with previous work on how students' self-efficacy shows positive correlation with students' online self-regulated learning while engaged in MOOC [32].

**Table 12.** SeQOL and OSRL correlations.

| | Correlation Matrix | | |
|---|---|---|---|
| | **OSRL Score** | **SeQOL Score** | **Grade** |
| OSRL Score | — | | |
| SeQOL Score | 0.572 *** | — | |
| Grade | 0.104 | 0.118 | — |

Note. *** *p* < 0.001.

#### 5.2.2. Correlations between Online Self-Efficacy and Learning Performances

Pearson correlation, shown in Table 12, showed no significant correlation between academic achievement and online learning self-efficacy (*r* = 0.118). These findings differ from previous works that measured the correlation between self-efficacy and learning performance (e.g., [33,34]). These previous works showed a positive and significant correlation between self-efficacy and learning performance.

#### 5.2.3. Correlations between Online Self-Regulated Learning and Learning performance

Pearson correlation, shown in Table 12, showed no significant correlation between academic achievement and self-regulated learning skills (*r* = 0.104). Our research findings differ from previous works that measured the correlation between self-regulated learning skills and learning performance (e.g., [9,35]). These previous works showed a positive and significant correlation between self-regulated learning component (i.e., learning goal) and learning performance. Interestingly, our research findings are similar to [9], in that no significant correlations were found between any of the other SRL components and learning performance: environment structuring; task strategies; time management; help seeking; and self-evaluation.

### 5.2.4. Differences in Learning Performance, OSRL and SeQOL

The results from independent *t*-test (Table 13) showed that there were no significant differences in the grades, OSRL, and SeQOL between male and female students, although generally male students had higher grades (Table 14).

**Table 13.** Independent sample *t*-test grouped by gender.

| | Independent Samples *t*-Test | | | |
|---|---|---|---|---|
| | | Statistic | *df* | *p* |
| SeQOL Score | Student's t | −0.932 | 290 | 0.352 |
| OSRL Score | Student's t | −0.171 | 290 | 0.864 |
| Grade | Student's t | −0.680 | 236 | 0.497 |

**Table 14.** Descriptive statistics between gender and grade, OSRL, and SeQOL.

| | Group Descriptives | | | | | |
|---|---|---|---|---|---|---|
| | Group | N | Mean | Median | SD | SE |
| SeQOL | Female | 161 | 160.1 | 169.0 | 51.8 | 4.08 |
| Score | Male | 131 | 165.9 | 180.0 | 52.7 | 4.60 |
| OSRL | Female | 161 | 108.1 | 111.0 | 22.4 | 1.77 |
| Score | Male | 131 | 108.6 | 112.0 | 22.8 | 1.99 |
| Grade | Female | 129 | 71.8 | 80.0 | 24.8 | 2.18 |
| | Male | 109 | 74.0 | 84.0 | 25.3 | 2.42 |

The results from independent *t*-test (Table 15) showed that there were no significant differences in the OSRL and SeQOL between age group (under 30 years old and 30+ years old), but there was a significant difference in the final grades between age group. The grades of students under 30 years old were significantly higher than the grades of students who were 30+ years old. Generally, older students (30+ years old) had higher scores in both OSRL and SeQOL, but lower grades (Table 16).

**Table 15.** Independent sample *t*-test grouped by age.

| | Independent Samples *t*-Test | | | |
|---|---|---|---|---|
| | | Statistic | df | *p* |
| OSRL Score | Student's t | −0.842 | 293 | 0.400 |
| SeQOL Score | Student's t | −2.075 | 293 | 0.039 |
| Grade | Student's t | 1.704 [a] | 237 | 0.090 |

[a] Levene's test is significant ($p < 0.05$), suggesting a violation of the assumption of equal variances.

**Table 16.** Descriptive statistic of grade, SeQOL, and OSRL grouped by age.

| | Group Descriptives | | | | | |
|---|---|---|---|---|---|---|
| | Group | N | Mean | Median | SD | SE |
| OSRL | under 30 years | 222 | 107.5 | 111.0 | 23.6 | 1.58 |
| Score | 30+ years | 73 | 110.1 | 111.0 | 19.4 | 2.27 |
| SeQOL | under 30 years | 222 | 159.3 | 171.5 | 52.8 | 3.54 |
| Score | 30+ years | 73 | 173.8 | 185.0 | 48.7 | 5.71 |
| Grade | under 30 years | 178 | 74.4 | 82.0 | 23.9 | 1.79 |
| | 30+ years | 61 | 68.2 | 80.0 | 27.5 | 3.52 |

### 5.3. Addressing RQ 3: Differences in Students' Levels of Online Self-Efficacy, Online Self-Regulated Learning, and Learning Performance by Demographic Variables

From the demographic data, we can conclude that students who accessed the Internet for 3–6 h achieved the highest grades. Self-efficacy in online learning tends to increase as



Internet usage is higher. Online self-regulated learning is higher when students use the Internet 3–6 h (Table 17).

**Table 17.** Descriptive statistic of grade, SeQOL, and OSRL grouped by Internet time usage.

|  | Internet Time | Grade | SeQOL Score | OSRL Score |
|---|---|---|---|---|
| Mean | 1–3 h | 70.9 | 126 | 96.7 |
|  | 3–6 h | 73.4 | 158 | 110 |
|  | >6 h | 72.9 | 170 | 109 |

The results from independent *t*-test (Table 18) showed that there were no significant differences in the OSRL and SeQOL between student group (social sciences and sciences and technology major). However, social science students generally had higher grades and OSRL and SeQOL scores (Table 19).

**Table 18.** Independent sample *t*-test grouped by major (social sciences and sciences and technology major).

| Independent Samples *t*-Test | | | | |
|---|---|---|---|---|
|  |  | Statistic | *df* | *p* |
| Grade | Student's t | 1.00 | 229 | 0.316 |
| SeQOL Score | Student's t | 1.29 | 281 | 0.197 |
| OSRL Score | Student's t | 1.57 | 281 | 0.117 |

**Table 19.** Descriptive statistic of grade, SeQOL, and OSRL grouped by major (social sciences and sciences and technology major).

| Group Descriptives | | | | | | |
|---|---|---|---|---|---|---|
|  | Group | N | Mean | Median | SD | SE |
| Grade | Social sci | 225 | 73.0 | 82.0 | 25.4 | 1.69 |
|  | Sci and tech | 6 | 62.5 | 69.5 | 19.7 | 8.04 |
| SeQOL Score | Social sci | 276 | 163.2 | 174.5 | 52.4 | 3.15 |
|  | Sci and tech | 7 | 137.3 | 169.0 | 51.8 | 19.56 |
| OSRL Score | Social sci | 276 | 108.8 | 112.0 | 22.8 | 1.37 |
|  | Sci and tech | 7 | 95.1 | 98.0 | 17.8 | 6.72 |

The results from independent *t*-test (Table 20) showed that there were no significant differences in the OSRL and SeQOL between semester group (students in 1st to 4th semester are labeled as junior and students in 5th to 8th semesters are labeled as senior), but there was a significant difference in the final grades between semester group. The juniors' grades were significantly higher than the seniors' grades. Generally, the junior's students had higher grades, OSRL, and SeQOL (Table 21).

**Table 20.** Independent sample *t*-test grouped by semester (junior and senior).

| Independent Samples *t*-Test | | | | |
|---|---|---|---|---|
|  |  | Statistic | *df* | *p* |
| Grade | Student's t | 2.519 [a] | 93.0 | 0.013 |
| SeQOL Score | Student's t | 0.587 | 113.0 | 0.558 |
| OSRL Score | Student's t | 0.575 | 113.0 | 0.567 |

[a] Levene's test is significant ($p < 0.05$), suggesting a violation of the assumption of equal variances.

**Table 21.** Descriptive statistic of grade, SeQOL, and OSRL grouped by semester (1–4 = junior and 5–8 = senior).

| | Group Descriptives | | | | | |
|---|---|---|---|---|---|---|
| | **Group** | **N** | **Mean** | **Median** | **SD** | **SE** |
| Grade | junior | 38 | 76.8 | 81.0 | 19.9 | 3.24 |
| | senior | 57 | 63.6 | 77.0 | 28.0 | 3.70 |
| SeQOL | junior | 45 | 176.5 | 188.0 | 49.7 | 7.40 |
| Score | senior | 70 | 171.2 | 177.5 | 46.4 | 5.55 |
| OSRL | junior | 45 | 113.7 | 116.0 | 18.2 | 2.71 |
| Score | senior | 70 | 111.5 | 113.0 | 21.7 | 2.59 |

*5.4. Addressing RQ 4: The Students' Learning Experiences, Strategies, and Challenges Faced while Engaged in Online Distance Learning Activities*

Qualitative data were gathered to describe students' learning experiences in five categories, including: (1) personal goals, (2) overall learning experiences, (3) learning challenges, (4) learning strategies, and (5) need for learning support.

(1)    Personal goals

Goal setting is a core component of self-regulated learning model [36,37]. Six themes related to personal goals were found (see Table 22): increase knowledge, insight, and abilities; enhance management knowledge at work; independent life goals; understand and master the lecture material; improve communication skills; and complete studies for self-development. Examples of the respondents' statements related to their personal goals can be read in Table 22.

**Table 22.** Statements related to personal goals.

| Personal Goals: Examples of the Respondents' Statements |
|---|
| "Understand all the courses, so that I can get satisfactory results or grades." |
| "Hope that the values taught in the learning materials can be applied in everyday life..." |
| "Gaining new knowledge and being able to provide the greatest benefit to others..." |
| "Obtain a degree..." |
| "Able to study independently online..." |

(2)    Learning experiences

Nine themes were revealed to be related to students' memorable learning experiences (see Table 23): the course was easy to understand; active and independent learning was implemented; impressive lecture materials; good learning environment; fun online experience; facilitated class discussions; flexible time to study; value-related experience; and organized online tasks and work. These themes echoed the learning experiences reported by Philipinal [38] and the Kingdom of Saudi Arabic students [39]. Examples of the respondents' statements related to learning experiences can be read in Table 23.

**Table 23.** Statements related to learning experiences.

| Learning Experiences: Examples of the Respondents' Statements |
|---|
| "The positive thing is that I can learn to discuss and share the problems of difficulties faced in following online lecture materials with other students in the same class of one major, I can also evaluate myself in facing end-of-semester exams, assignments." |
| "I can do tasks anywhere according to the time that has been set. I am flexible in managing time." |
| "When I study alone using the module I don't really understand it; but when we discuss and exchange ideas online, it makes me understand what I'm learning..." |

(3)    Learning challenges

Ten themes related to challenges were found (see Table 24): independence needed, sometimes there are distractions from work, bandwidth/signal, searching for information, difficult course material, source material, online exams, learning while working, daily activities, and friend factor as a distractor. These themes confirmed findings from previous studies [40,41]. Examples of the respondents' statements related to learning challenges can be read in Table 24.

**Table 24.** Statements related to learning challenges.

| Learning Challenges: Examples of the Respondents' Statements |
| --- |
| "Time challenge because I am a factory worker who can only do assignments when I get home from work and have to give up bedtime to stay up late." |
| "It is just that internet connection must be maintained, so that there are no obstacles in learning online." |
| "The challenge is when we do not fully understand the material but the task continues." |

(4)    Learning strategies

Nine themes related to learning strategies were found, including general study behaviors and self-regulated learning behaviors: follow online lecture materials and reading materials; listen to information from the lecturer; look for a study schedule outside of working hours; read to the end; optimize discussion time; read at night; take advantage of empty sessions; read modules and complete assignments; and set study time. Examples of the respondents' statements related to their learning strategies can be read in Table 25. Students in online learning programs need to apply self-regulated learning skills, which were reported to have a positive relationships with learning outcomes (e.g., [8,9]).

**Table 25.** Statements related to learning strategies.

| Learning Strategies: Examples of the Respondents' Statements |
| --- |
| "First, read and understand each chapter that will be studied in each session, then take notes." |
| "The strategy I use is good time management, especially since I am a worker. I need to schedule when to study and work on assignments from lecturers so as not to experience problems with my work." |
| "Being active in sharing learning information either by reading modules or accessing the courses studied through the learning system." |
| "When I study I will look for a place that makes me focus." |

(5)    Suggestions for learning support

Ten themes related to suggestions were found regarding various aspects of learning support that needed improvement: tutor feedback; input related to learning materials; input related to discussion responses; suggestions for learning improvement; lecturer–student interaction; online lectures; system maintenance; lecturers' responses to help seeking; stability of infrastructure; and synchronous learning sessions. Examples of the respondents' statements related to suggestions for learning support can be read in Table 26. Such needs for support can be addressed by facilitating students' self-regulated learning, metacognition, and help-seeking [31,42,43], which we explore more in next section.

**Table 26.** Statements related to suggestions for learning support.

| Suggestions for Learning Support: Examples of the Respondents' Statements |
|---|
| "Lecturers could be more active in responding to students."<br>"I hope to build communication with classmates."<br>"The University should provide convenience in achieving final results and give us more opportunity to discuss assignments, so that students can better understand the assignments."<br>"Maybe more videos from lecturers. Because it will be helpful for students to understand learning materials." |

## 6. Discussion and Recommendations

The quantitative results revealed that IOU students' online self-efficacy and online self-regulated learning were above average. However, students reported low levels of confidence and ability to seek help from others, and low levels of abilities to develop task strategy and allocate time for learning online, which are the weak areas in which they should improve themselves. Further, online learning self-efficacy and OSRL were positively and significantly correlated, which supports previous research evidence (e.g., [16]). However, there was no significant correlation between online learning self-efficacy and learning performance and between online self-efficacy and learning performance, which was somewhat surprising given that previous studies reported positive relationships among these variables [8,9]. The insignificant relationships may be due to the fact that students' course assessments did not require them to possess abilities to self-manage and have confidence to deal with all aspects of online learning that were examined by the OSRL and SeQOL questionnaires.

Our results revealed equity issues that require online distance learning educators' attention. First, male and younger students obtained significantly higher grades than female and older students. Second, students enrolled later in the online learning programs (students studying in 1st to 4th semester) reported higher levels of online self-efficacy, online self-regulated learning, and learning performance than students who enrolled earlier (students studying in 5th to 8th semester), which might require attention by lecturers of the latter student cohorts. Therefore, lecturers need to understand the learning needs among female students, students who are older, and students taking different courses to provide appropriate learning support. Third, an interesting result was that students who spent 3–6 h accessing the Internet per week had higher learning performances than those spending less or more time on the Internet. This result suggests that spending the right amount of time might make online learning more productive, which is related to time management in self-regulated learning online.

The qualitative findings provided descriptive information on students' learning goals, experiences, challenges, strategies, and need for support, which helped us interpret the quantitative results. For example, students had difficulties in asking their classmates questions and obtaining feedback from their lecturers, which echoed their lack of confidence and ability to seek help from others.

To address the above-mentioned areas for improvement and equity issues for IOU students' online learning, we propose the following recommendations for key stakeholders of IOU and other open universities in similar contexts. First, students need to pay particular attention to three areas of self-regulation of online learning for improvement that are identified in this study. These include: (1) improving time management in reading learning materials and completing assignments, (2) developing task strategies for completing assignments, and (3) seeking help from classmates during online learning interaction. Each of these areas are important for students' success in online learning [31]. Students should explore suitable strategies for managing limited study time by capitalizing on the flexibility of online learning tools and materials, identifying learning strategies that are suitable to them, and being proactive in interacting with peers in the online learning environment.

Second, lecturers are recommended to offer training workshops and interactive self-learning materials, and engage students in collaborative learning activities, which can

provide a dynamic online learning environment to offer students responsive guidance and feedback that help them cope with learning challenges. This is supported by the Community of Inquiry model that suggests lecturers to engage students in frequent social interaction and productive inquiry through activities such as online discussions [42,44,45].

Third, the open university management needs to provide lecturers with professional development regarding strategies to promote students' online self-regulated learning and self-efficacy for online learning [44]. Meanwhile, the open universities should promote lecturers' applied research initiatives, such as using learning analytics to predict success and failure early, providing timely learning support, and designing visualization tools to make students' self-regulated learning behaviors visible to themselves and lecturers to better support their online learning [46].

Fourth, information technology provision should be guaranteed. The students reported technical problems, such as difficulties in using online learning systems, poor Internet connection, and limited support by technical staff support. These problems can be resolved through improving the technological infrastructure and capacity building for technical staff with a view to supporting students' effective online learning at a distance.

## 7. Conclusions

This paper reports one of the first studies focusing on Indonesian online distance learning students' self-regulated learning. Some of the results echoed previous research evidence, such as the significant relationship between online learning self-efficacy and online self-regulated learning. Other results were inconsistent with existing evidence, such as the insignificant relationships of these online learning self-efficacy and online self-regulated learning with students' learning performance. Although students had above-average online learning self-efficacy and online self-regulated learning, they also reported a lack of abilities and/or confidence in formulating task strategies, allocating time for online learning, and seeking help from others. Further, equity issues emerged in relation to the lower learning performance of female and older students compared with male and younger students. The above findings alert lecturers and management of open universities to provide suitable learning support to help students improve self-regulated learning skills with a view to supporting all students' success in online distance learning.

The aforementioned results warrant future research using different methodologies to investigate the perspectives of multiple stakeholders in different online distance education contexts. Although the results of this study cannot be generalized to the whole student population at IOU because the participants did not come from all study programs, the recommendations based on the quantitative and qualitative analysis results might be considered by teaching, technical staff, and management at the IOU and other open universities in providing learning support.

**Author Contributions:** Conceptualization, H.B.S., R.D.R., T.P, F.A.T.H.S., A.S. and M.Y.; Formal analysis, H.B.S., R.D.R., F.A.T.H.S. and M.Y.; Investigation, H.B.S., R.D.R., T.P, F.A.T.H.S. and A.S.; Methodology, H.B.S., R.D.R., T.P, F.A.T.H.S., A.S. and M.Y.; Project administration, R.D.R.; Supervision, H.B.S. and M.Y.; Writing—original draft H.B.S., R.D.R., T.P, F.A.T.H.S., A.S. and M.Y.; Writing—review & editing, H.B.S. and M.Y. All authors have read and agreed to the published version of the manuscript.

**Funding:** This research was funded by The Open University of Indonesia, grant number 21661/UN31.LPPM/PT.01.03/2021.

**Institutional Review Board Statement:** This research was funded by The Open University of Indonesia grant number: 21661/UN31.LPPM/PT.01.03/2021.

**Informed Consent Statement:** Informed consent was obtained from all subjects involved in the study.

**Conflicts of Interest:** The authors declare no conflict of interest.

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
