# Peer review of "Learners’ Online Self-Regulated Learning Skills in Indonesia Open University: Implications for Policies and Practice"

_education, doi:10.3390/educsci12070469_

Round 1

Reviewer 1 Report

This study contributes to the understanding SRL in online learning from Indonesia students. A few suggestions for improvements:

1. the demographics variables (age, gender, etc.) of the survey should be indicated in the methods section. It would be easier for readers. I don't see the variables until reading the results section. 

2. Under "data analysis", for quantitative data, only Person analysis approach was indicated. There are other analyses that were performed. Please indicated them as well.

3. In the results section, authors probably could just report descriptive info for each dimension for both SRL and self-efficacy scales. But if the authors prefer to report the info for each item, it would be fine.

Author Response

Responses to Reviewer 1’s comments:

Comment 1

The demographics variables (age, gender, etc.) of the survey should be indicated in the methods section. It would be easier for readers. I don't see the variables until reading the results section.

Response 1

The authors have added the demographics variables (age, gender, etc) of the survey in the methods section.
We have “Participants and Procedures” subsection under the Method section that describes the demographics information.

Comment 2

Under "data analysis", for quantitative data, only Person analysis approach was indicated. There are other analyses that were performed. Please indicated them as well.

Response 2

The authors have performed other analyses; not only Pearson analysis.
The section has been updated by addressing the reviewer’s suggestion.

Comment 3

In the results section, authors probably could just report descriptive info for each dimension for both SRL and self-efficacy scales. But if the authors prefer to report the info for each item, it would be fine.

Response 3

The authors prefer to report the information for each item.

Reviewer 2 Report

The overall manuscript is neat and written concisely—with relevant information for existing literature. One aspect that you can focus on is the coherence in your work. Some sections are fragmented. In addition, the way you phrase things might need improvement (see below) as well the presentation of your results. Moreover, there are some inconsistencies in displaying information (commas, etc.).

Author Response

Responses to Reviewer 2’s comments

Comment 1:

p.abstract

After reading the abstract, I have the feeling that you add the word “online” to every concept you introduce. Research does not work like that. For example, in the sentence “adopt or fail to adopt online self-regulated learning to engage in online distance learning” the first “online” is redundant. You explain how to implement those self-regulated learning strategies to an online setting. In addition, you have to review whether online plus distance is necessary. I think “online learning” is suffice; however, I do not know if this is the message you intend to convey.

Response 1:

Thank you very much for the suggestion. The authors have made adjustment to address this suggestion.

Comment 2:

p.abstract

“female and older students” → Do you mean: female and/or older students?

Response 2:

Has been updated.

Comment 3:

p.abstract

Why do you target open universities? Do you exclude “regular” universities?

Response 3:

This study targeted Open Universities with Indonesia Open University as a case study. Open Universities have long experiences in delivering (regular or traditional) distance education compared to the “regular’ universities. Many studies have been conducted in the context of traditional distance education. However, there are still limited studies in investigating the online context of distance education, more specifically their students’ online self-efficacy and online self-regulated learning.  

Comment 4:

p.abstract

What are those “similar context” you are talking about?

Response 4:

We are talking about similar context with online distance learning such as online learning, web-enhanced learning, and MOOC.

Comment 5:

Why do you include “Indonesia Open University” as a keyword? It would make more sense to use “open university/universities”.

Response 5:

The authors have edited the keyword and used “open university”.

Comment 6:

p.1 There is a lack of coherence on this page. The sentences do not follow one another logically.

Response 6:

Has been updated.

Comment 7:

p.1 You also present back-to-back brackets: (x)(y). Avoid this. Moreover, you talk about online learning environments, but how does that relate to digital and/or virtual environments?

Response 7:

Has been updated to avoid back-to-back brackets

According to our understanding these are similar terms. However, not all digital environments are online learning environments.

Comment 8:

p.2 Text is presented in a smaller font. In addition, you present a quote from Bandura (1997); however, you do not mention the page number. Please insert this. Again you have back-to-back brackets. Plus if you insert author names (three or more), you can immediately write first author + et al. You can apply this throughout your manuscript.

Response 8:

Has been updated.

Comment 9:

p.2 The description “traditional university students” together with “k-12 students” is odd. They are two different types of participants. First you focus on the age; then on the location (i.e., on-site).

Response 9:

Have been updated.

Comment 10:

p.3 The sentence below “Methods” is not a paragraph (a paragraph is at least three sentences). In addition, you mention the number of participants. This should be inserted with spaces before and after the “=”.

Response 10:

Have been updated.

Comment 11:

p.table1 You use a comma in the first paragraph on this page whereas you use periods in the table (and in the second paragraph). This should be consistent.

Response 11:

Have been updated.

Comment 12:

p.3 The scale with 11 response options is an 11-poin Likert-scale. I would suggest to read more research in this area to familiarize yourself with more frequently used terms.

Response 12:

Have been updated.

Comment 13:

p.4 I would suggest to make Table 4 smaller (or embed the information in the main text). Please check if the text in the tables is a different font colour. Moreover, you place the options from the Likert-scale (again presented as “6 response options”), in italics. You do not do this earlier in your work. Check the remainder of your work to make this consistent.

Response 13:

Have been updated.

Comment 14:

p.5 You mention: “These are…learning support”. The questions itself do not ask something; you do that. Please use the correct verb in this sentence. Moreover, in the following paragraph you mention ”correlation among online selfefficacy”. It is the correlation between. In the last sentence of that paragraph: the “also” is redundant.

Response 14:

Have been updated.

Comment 15:

p.5 “2 or more devices” = two or more devices.

Response 15:

Have been updated.

Comment 16:

p.5 “but lower in final grade” = “but a lower final grade”.

Response 16:

Has been updated.

Comment 17:

p.6 In the tables, replace “number” with “frequency”.

Response 17:

Has been updated.

Comment 18:

p.6 Are the page numbers displayed in a different font? Please check.

Response 18:

Has been updated.

Comment 19:

p.6 Again the “11 response options” = 11-point Likert-scale.

Response 19:

Has been updated.

Comment 20:

p.7 Is it possible to display the table horizontal so the statements are on one row? I think that will make the table easier to read.

Response 20:

We need help from the technical editor to accommodate this suggestion.

Comment 21:

p.7 I would avoid the unnecessary repetition. You present information in a lengthy Table. Therefore, it is unnecessary to repeat it in the main text as well. Moreover, the last sentence on that page belongs in the conclusion (not in the results).

Response 21:

Has been updated.

Comment 22:

p.9 Insert spaces surrounding the statistics in the main text: “M=4.08” = “M = 4.08”. If you delete the redundant sections, displaying your statistics this way should not increase the max. number of words.

Response 22:

Has been updated.

Comment 23:

p.10 Table 13 has a different set-up than the table earlier (with similar information). The headers on this page do not follow a referencing format (such as APA). Please check and adjust accordingly.

Response 23:

Has been updated.

Comment 24:

p.11 There is an additional horizontal row in the table below “OSRL Score” that is redundant.

Response 24:

Has been updated.

Comment 25:

p.13 Why do you place these themes in a Table (and in italics + boldface)?

Response 25:

To highlight the message and hopefully decrease the use of the paper space.

Comment 26:

p.15 “provided rich information” → Such as?

Response 26:

They are information on students’ learning goals, experiences, challenges, strategies, and need for support, which helped us interpret the quantitative results.

We thought it is a good idea to change the word with “descriptive information” instead of “rich information” since the idea is to provide additional information for the quantitative analysis.

Comment 27:

p.15 The summary (i.e., First, Second, Third) should not be placed in italics. In addition, the first paragraph should be merged with “To address the…in similar contexts”.

Response 27:

Have been updated by following the suggestions.

Comment 28:

p.references

You need to go over the following aspects in your reference list to make things more consistent: (a) hyphens, (b) Oxford comma, and (c) doi links (it should contain the https either with or without the hyperlink).

Response 28:

Have been updated by following the suggestions.